# Estimating Crop Biophysical Parameters Using Machine Learning Algorithms and Sentinel-2 Imagery

**Mahlatse Kganyago** [1,2,*] **, Paidamwoyo Mhangara** [1] **and Clement Adjorlolo** [1,3]

1 School of Geography, Archaeology and Environmental Studies, University of the Witwatersrand, Johannesburg 2050, South Africa; paida.mhangara@wits.ac.za (P.M.); clementa@nepad.org (C.A.)
2 Earth Observation, South African National Space Agency, The Enterprise Building, Mark Shuttleworth Street, Pretoria 0001, South Africa
3 African Union Development Agency (AUDA-NEPAD), 230 15th Rd, Midrand 1685, South Africa
* Correspondence: mkganyago@sansa.org.za; Tel.: +27-(0)12-844-0424

**Abstract:** Global food security is critical to eliminating hunger and malnutrition. In the changing climate, farmers in developing countries must adopt technologies and farming practices such as precision agriculture (PA). PA-based approaches enable farmers to cope with frequent and intensified droughts and heatwaves, optimising yields, increasing efficiencies, and reducing operational costs. Biophysical parameters such as Leaf Area Index (LAI), Leaf Chlorophyll Content ($LC_{ab}$), and Canopy Chlorophyll Content (CCC) are essential for characterising field-level spatial variability and thus are necessary for enabling variable rate application technologies, precision irrigation, and crop monitoring. Moreover, robust machine learning algorithms offer prospects for improving the estimation of biophysical parameters due to their capability to deal with non-linear data, small samples, and noisy variables. This study compared the predictive performance of sparse Partial Least Squares (sPLS), Random Forest (RF), and Gradient Boosting Machines (GBM) for estimating LAI, $LC_{ab}$, and CCC with Sentinel-2 imagery in Bothaville, South Africa and identified, using variable importance measures, the most influential bands for estimating crop biophysical parameters. The results showed that RF was superior in estimating all three biophysical parameters, followed by GBM which was better in estimating LAI and CCC, but not $LC_{ab}$, where sPLS was relatively better. Since all biophysical parameters could be achieved with RF, it can be considered a good contender for operationalisation. Overall, the findings in this study are significant for future biophysical product development using RF to reduce reliance on many algorithms for specific parameters, thus facilitating the rapid extraction of actionable information to support PA and crop monitoring activities.

**Keywords:** leaf area index; leaf chlorophyll content; canopy chlorophyll content; Sentinel-2; sparse partial least squares; random forest; Gradient Boosting Machine

## 1. Introduction

Globally, achieving food security is one of the most significant challenges facing humanity [1]. Addressing such a colossal challenge will require global food production to be increased by 70% to feed an estimated 9 billion people by 2050 [2]. In addition, the capacity of agricultural systems to sustain food security is undermined by climate change and associated disasters such as droughts and heatwaves. These climatic phenomena lead to reduced agricultural systems productivity and loss of profits and livelihoods. To cope with the climatic variability and change, farmers must adopt resilient technologies and farming practices such as precision agriculture to increase efficiencies and productivity and reduce operational costs. Precision agriculture (PA) is an innovative management practice that utilises a set of technologies and principles for site-specific management of spatiotemporal variability related to various aspects of agricultural production [3]. As a food-insecure region, Africa needs greater adoption of PA to address food security challenges and ensure the resilience and profitability of agro-ecological systems. For

example, variable rate application (VRA) of fertilisers, herbicides, and pesticides, i.e., where and when needed, will reduce their loading in the environment and management costs relative to the traditional uniform application [4,5]. For instance, better wheat grain quality [6], reduced pesticide cost by 60–67% and time, and fuel and labour for application by 27–32% and 28%, respectively, have been reported [5]. Moreover, Stamatiadis, et al. [4] show the reduced effects of excess nitrogen (N) application such as soil acidity, electrical conductivity, and residual nitrates in the root zone of cotton.

Remote sensing satellites are critical for the rapid assessment of crop health and the development of spatially explicit site-specific information to facilitate VRA technologies and field management. Therefore, they are one of PA's most promising and sustainable technologies due to their well-established capability to acquire data over large areas remotely and repeatedly at various spatial and temporal resolutions. One of the most established approaches to assess crop spatial variability and monitor agricultural systems' productivity is by estimating key crop biophysical parameters such as Leaf Area Index (LAI), Leaf Chlorophyll Content ($LC_{ab}$), and Canopy Chlorophyll Content (CCC). Leaf area index (LAI)—defined as the one-sided area ($m^2$) of the total developed green leaf area per unit ground surface area ($m^2$) in broadleaf canopies [7]—is an essential biophysical parameter that provides valuable information on plant physical and physiological processes, and thus is critical for characterising crop growth status and health and stress.

In contrast, chlorophyll is the main biochemical parameter involved in photosynthesis and related to nitrogen (N) content [8]. Therefore, leaf and canopy chlorophyll content ($LC_{ab}$ and CCC) are essential parameters for determining photosynthetic capacity, optimising N application to increase yields and profits, and reducing environmental impact from excessive fertilization [9]. Traditionally, these parameters have been measured using direct lab-based methods, which are destructive, spatially, and temporally limited, expensive, time-consuming, and labour-intensive. Therefore, remotely sensed approaches for biophysical parameter estimation are highly sought to rapidly and frequently characterise crop conditions at regional scales to inform the development of policy instruments for alleviating the impacts of climate change. On a landscape scale, they facilitate land management decision-making by providing information about when and where to fertilize or irrigate and by how much, thus promoting the sustainability and profitability of agricultural systems.

Remote sensing satellites measure the varying reflectance properties determined by canopy biophysical and biochemical parameters [10]. For example, reflectance properties in the visible region (VIS, 350 nm–649 nm), characterised by intense absorption in the blue (350 nm–449 nm) and red (550 nm–649 nm) bands, are caused by carotenoids, xanthophyll, anthocyanin, and chlorophyll pigments [11]. The near-infrared (NIR, 750 nm–1299 nm) region is characterised by the high reflectance caused by the spongy mesophyll cells, canopy structure, and water content in leaves [10], while the plant canopy reflectance in the SWIR region (1300 nm–2500 nm) is affected by subtle biochemical properties such as sugars, cellulose, starch, protein, and water content [12]. Recent studies [13,14] suggest that the canopy spectral properties are influenced by the interaction of various biophysical parameters. Accordingly, the VIS-NIR and SWIR regions are commonly used for estimating crop biophysical parameters using parametric statistical approaches such as spectral vegetation indices (SVIs), non-parametric techniques such as Machine Learning Regression Algorithms (MLRAs), inversion of coupled leaf-canopy radiative transfer models (RTMs) such as PROSAIL [15,16], and hybrid approaches such as RTMs and MLRAs [17]. Critically, all parametric approaches, i.e., SVIs, regardless of their form—simple ratios, spectral band differences, and normalised difference ratios—are sensitive to the changes in soil background at early growth stages, leaf chlorophyll content, and saturate at lower and higher canopy cover, i.e., LAI < 2 and >5 [17,18].

Commercial sensors such as Rapid-Eye and Worldview-2 have improved capabilities and prospects for estimating biophysical parameters by including narrow red-edge (RE) bands [19]. The RE spectral region (650 nm–749 nm) is characterised by a steep increase in

vegetation reflectance from strong chlorophyll absorption in the red band to high foliage reflection in the NIR, i.e., LAI. This region is sensitive to changes in the leaf chlorophyll content and plant stress and has been shown to improve biophysical parameters estimation accuracy [20,21]. Previously, exorbitant costs of commercial image data limited the full potential and exploitation of the RE region [22]. The advent of Sentinel-2 Multi-Spectral Imager (MSI), with three new RE bands centred at 705 nm, 740 nm, and 783 nm, is particularly attractive due to its free access and offers prospects of high spatial (i.e., <20 m) and temporal resolution (i.e., five days) LAI, $LC_{ab}$ and CCC for precision agriculture and crop monitoring applications [23]. Using the Sentinel-2 and Sentinel-3 band settings, Clevers and Gitelson [24] observed that the RE indices, i.e., red-edge chlorophyll index (CIred-edge), the green chlorophyll index (CIgreen), and the MERIS terrestrial chlorophyll index (MTCI), were linearly related to CCC and nitrogen (N) content and hence were accurate estimators. However, Sakamoto, et al. [25] argue that SVIs are not transferable over various canopy architectures, leaf structures, climate zones, and environmental conditions and contain lower information content.

Alternatively, physically-based methods such as PROSAIL [15] can simulate spectral properties of complex vegetation canopies and solar and acquisition angles through RTMs. Physically-based methods were shown to be more robust, accurate, and transferable over various cover types and environments [26,27]. However, they require substantial site-specific data for parameterisation, which has proven difficult, costly, and considerably slow [17,20]. Furthermore, physically-based approaches are ill-posed (i.e., different combinations of canopy parameters may correspond to almost similar spectra, thus yielding inconsistent results), complex, and computationally expensive [27,28]. In contrast, non-parametric approaches, i.e., MLRAs, require less parameterisation, are relatively fast, and can learn non-linear relationships. Over the years, the MLRAs have evolved into reliable approaches, implementable at various spatial and temporal scales. The MLRAs such as random forest (RF) [29], support vector machines (SVM) [30], and artificial neural networks (NN) [31] were demonstrated to be more robust to noisy features, small training size, and high dimensionality (when $p > n$) and collinearity [20,32,33].

Recent studies show that the coupling of physically based approaches and MLRAs, i.e., hybrid approaches, yields better prediction accuracies. These approaches are available operationally, such as the Sentinel-2 Level-2 Prototype Processor (SL2P) through Sentinel Application Platform (SNAP) and Sen2Agri system [34,35] for PROBA-V, Sentinel-2 MSI, and Landsat-8 Operational Land Imager (OLI). Although some validation and intercomparison studies [36] show their robustness and better accuracies in various agricultural environments, others [37,38] show poor accuracies with RMSE > 1 m$^2$ m$^{-2}$. In contrast, simple and more robust MLRAs such as Random Forest [29], Gaussian Process Regression (GPR) [39], and Kernel Ridge Regression (KRR) [40] have better estimation accuracy and computational cost, can learn complex relationships from input data, and require few and intuitive tuning parameters [17,20,41]. Due to continuous improvements in MLRA, optimisation techniques (e.g., Gradient descent), sensors capabilities (such as increasing resolution and number), and evolving user needs [42], it is essential to evaluate various MLRAs to improve site-specific accuracy and reliability and inform future operationalisation, towards the improvement of food production in developing countries.

This paper assessed various MLRAs to estimate crop biophysical parameters using Sentinel-2 data in the Bothaville site characterised by medium- to large-commercial farming systems. The specific objectives were (1) to evaluate and compare the predictive performance of sparse Partial Least Squares (sPLS), Random Forest (RF), and Gradient Boosting Machine (GBM) in estimating LAI, $LC_{ab}$, and CCC using Sentinel-2 Multi-Spectral Imager (MSI) data, and (2) to identify the influential spectral bands with high predictive power for estimating the crop biophysical parameters. These algorithms have been limitedly exploited in the context of crop biophysical parameters estimation from Sentinel-2 imagery for precision agriculture and crop monitoring. Moreover, the comparison of the three MLRAs in the context of crop biophysical parameter estimation is worthwhile to elucidate

their capabilities under the same environmental and acquisition conditions and their consistency to previous performances as they may provide a promising alternative to existing operational techniques.

## 2. Materials and Methods

### 2.1. Study Area

This study was conducted in Bothaville, i.e., the main agricultural production zone of South Africa (Figure 1). The area is characterised by both commercial medium- to large-scale farming of crops and livestock. Bothaville is located at latitudes: 27°13′0″S to 28°8′0″S and longitudes: 26°0′0″E to 27°05′0″E. Characterised by warm and wet summers, the average temperature is ~18 °C, with an annual average rainfall of ~584 mm. Winters are cold, with average temperatures of about 5.1 °C. The dominant summer crops are maize, sunflower, and groundnuts, while winter crops include wheat and barley grown on sandy to sandy-loamy soils on generally flat slopes. The summer cropping calendar starts in December and ends in May or June, while the winter crop calendar starts in May and ends in October or November each year (https://ipad.fas.usda.gov/rssiws/al/crop_calendar/safrica.aspx, accessed on 5 June 2021). Bothaville is one of two sites used for testing, calibration, and validation of agricultural monitoring and early warning products within the context of the AfriCultuReS project (Enhancing Food Security in African AgriCultural Systems with the support of Remote Sensing, Grant Agreement No. 774652).

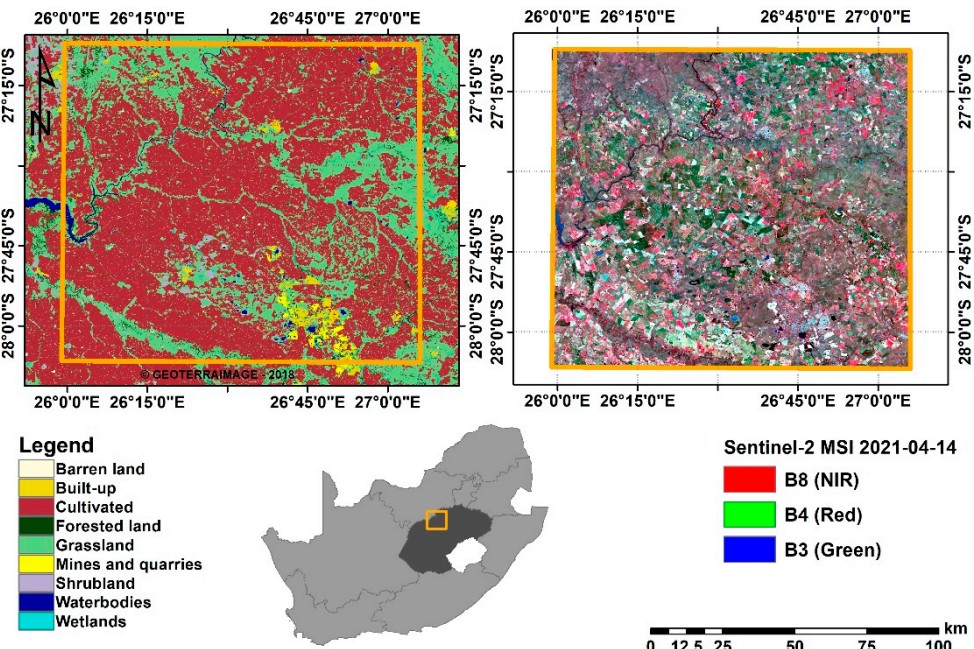

**Figure 1.** Location of Bothaville (Orange) in Free State province (Dark grey), South Africa. The panel on the left shows the land cover distribution in 2018 and the one on the right shows Sentinel-2 False-colour composite (Red: B8, Green: B4, and Blue: B3) acquired on the 14-04-2021.

### 2.2. Data

#### 2.2.1. Remotely Sensed Data

Sentinel-2 Multi-Spectral Imager (MSI) image acquired on the 14 April 2021 covering the study areas was downloaded from Sentinel Hub Cloud API for Satellite Imagery (Sinergise Laboratory for geographical information systems, Ltd., Ljubljana, Slovenia). These images are provided at Level-1C processing, i.e., Top-of-Atmosphere (TOA) reflectance in cartographic geometry. The MSI images consisted of 12 bands at three different spatial resolutions, i.e., 10 m (Band 2-Blue: 490 nm, Band 3-Green: 560 nm, Band 4-Red: 665 nm, and Band 8-NIR1: 842 nm), 20 m (Band 5-Red-edge: 705 nm, Band 6-Red-edge: 740 nm, Band 7-Red-edge: 783 nm, Band 8A-NIR2: 865 nm, Band 11-SWIR1: 1610 nm, and Band

12-SWIR2: 2190 nm), and 60 m (Band 1-Deep blue: 443 nm, Band 9-NIR3: 945 nm, and band 10-Cirrus: 1375 nm). For this study, all spectral bands at 10 m, 20 m, and 60 m resolution were used for the study, with 60 m bands dedicated to atmospheric corrections and cloud screening [43]. For estimation of crop biophysical parameters, we used 10 m and 20 m spectral bands to take advantage of the entire MSI spectral range. The Level-1C data were converted to Bottom-of-Atmosphere (BOA) reflectance, i.e., Level-2A, using Sen2Cor version 2.9 [44].

Sen2Cor is a processor for atmospheric correction (including cirrus clouds and terrain correction) of Sentinel-2 data. The algorithm converts the TOA (Level-1C) image data to BOA reflectance based on the libRadtran database of look-up tables (LUTs) generated for a wide variety of atmospheric conditions, solar geometries, and ground elevations. Scene classification is also performed as part of the processing to flag clouds, snow, and cloud shadows. Further details can be found in Mueller-Wilm [44] and Louis, et al. [45]. The image data was corrected using parameters: atmospheric model 'Mid-latitude summer', aerosol type 'Rural', and two-band water volume retrieval (i.e., 940 nm and 1130 nm). The image data were resampled to 20 m spatial resolution using the nearest neighbour resampling technique in Sentinel-2 Toolbox within SNAP software (SNAP-ESA Sentinel Application Platform v8.0, http://step.esa.int, accessed on 10 November 2020) due to its capability to preserve the fidelity of the pixel values.

### 2.2.2. Calibration and Validation Data

The data for calibrating and validating LAI, $LC_{ab}$, and CCC MLR models were collected in the field from 11 to 23 April 2021 over the dominant crops at the study area, i.e., Maize, Beans, and Peanuts. LAI and $LC_{ab}$ measurements were collected non-destructively within 40 m × 40 m plots to avoid edge effects and allow biophysical parameter mapping at 20 m resolution. The plots were selected systematically along a transect to capture variability. Each plot consisted of an average of six to eight random measurements. For LAI measurements, we used LiCor 2200c Plant Canopy Analyzer (Li-Cor, Inc., Lincoln, NE, USA), with a 180° view cap to shield the influence of the operator and unequal sky conditions on the measurements. Trimble® TDC600 handheld Data Collector, with global navigation satellite systems (GNSS) accuracy of 1.5 m, was used to Geo-tag the centroid of each plot and take plot pictures. In contrast, $LC_{ab}$ measurements were collected with MC-100 Chlorophyll Concentration Meter (Apogee Instruments, Inc., Logan, UT, USA) from the sunlit upper canopy. The Canopy Chlorophyll Content (CCC) for each plot was estimated as a product of $LC_{ab}$ and LAI ($LC_{ab}$ × LAI) [16]. The data were divided into 70% training and 30% validation [46]. The summary statistics for calibration and validation are given in Table 1.

**Table 1.** Descriptive statistics of the calibration (70%) and validation (30%) datasets for the measured LAI (m$^2$ m$^{-2}$), $LC_{ab}$ (µg cm$^{-2}$), and CCC (µg cm$^{-2}$).

|  | **Datasets** | *n* | **Min** | **Mean** | **Max** | **SD** |
|---|---|---|---|---|---|---|
| LAI | Calibration | 113 | 1.78 | 3.35 | 5.57 | 0.86 |
|  | Validation | 48 | 2.02 | 3.60 | 5.75 | 1 |
| $LC_{ab}$ | Calibration | 113 | 4.06 | 33.87 | 66.18 | 14.93 |
|  | Validation | 48 | 3.69 | 32.62 | 70.69 | 19.20 |
| CCC | Calibration | 113 | 10.30 | 105.69 | 288.22 | 61.60 |
|  | Validation | 48 | 7.87 | 116.42 | 339.10 | 90.39 |

### 2.3. Crop and Green-Vegetation Masking

Non-croplands were masked from the Sentinel-2 MSI data using a crop mask derived from the 2017 National Crop Boundaries Dataset [47] provided by the Department of Agriculture, Land Reform and Rural Development (DALRRD) of South Africa. This dataset is generated from SPOT 5 and SPOT 6 data and consists of all the active crop

fields (including small-holder and commercial fields) in a specific period, i.e., 2014–2015. Then, the fallow crop fields and non-vegetated pixels within these fields were removed from further analysis using a vegetation mask. The vegetation mask was created based on a Normalised Difference Vegetation Index (NDVI) with no vegetation threshold of <0.2, determined through trial and error (Equation (1)). In doing so, further analysis was constrained to the active crop fields in the 2021 growing season since the boundaries did not necessarily represent the current situation in the study area, i.e., at the period of fieldwork.

$$\text{Vegetation mask} = \begin{cases} \text{NDVI} < 0.20 = 0 \\ \text{NDVI} > 0.21 = 1 \end{cases} \tag{1}$$

*2.4. Machine Learning Regression Algorithms*

The selection of Machine Learning Regression Algorithms (MLRAs) considered here was based on their simplicity, quick training, robust performance, and popularity in various remote sensing applications. For example, the sparse Partial Least Squares regression (sPLS) algorithm has been widely used in remote sensing applications, e.g., crop yield prediction [48], grass biomass, and LAI estimation [49,50], while Random Forest (RF) has become one of the most popular algorithms for biophysical parameter estimation using a variety of remotely sensed data [32,51,52]. On the other hand, GBM is computationally effective and often outperforms other algorithms [53,54]. The three are also regarded interpretable, i.e., essential for obtaining novel insights (e.g., new causal links between explanatory and response variables) and troubleshooting the models (i.e., detecting and diagnosing biases in the input data and trained models) for improved satellite-based value-added products [55] as opposed to kernel-based and deep learning algorithms such as Kernel Ridge Regression (KRR), Support Vector Machines (SVM), and Neural Networks (NN), which are "black-boxes", complex, and computationally expensive [56]. Therefore, the comparison of these three MLRAs in the context of crop biophysical parameter estimation is worthwhile to elucidate their capabilities under the same environmental and acquisition conditions and consistency to previous performances. In addition, they may provide a promising alternative to existing operational techniques.

2.4.1. Sparse Partial Least Squares

The sparse Partial Least Squares regression (sPLS) [57] is the improvement of ordinary PLS [58] to perform variable selection and prediction simultaneously, thus it is useful for reducing multicollinearity and the phenomenon so-called the "curse of dimensionality" (i.e., $p > n$) [59]. Chun and Keleş [57] achieved this by promoting sparsity using $L_1$ the penalty imposed at the dimension reduction step of ordinary PLS, which promote exact zero property on the surrogate of direction vector $c$ rather than original direction vector $\alpha$ but keep both direction vectors close to each other. Thus, sPLS objective function, by Chun and Keleş [57], is given by Equation (2).

$$\min_{\alpha, \beth} -\kappa \alpha^T M \alpha + (1-\kappa)(c-\alpha)^T M(c-\alpha) + \lambda_1 |c|_1 + \lambda_2 |c|_2,$$
$$\text{subject to } \alpha^T \alpha = 1. \tag{2}$$

where $M = X^T Y Y^T X$. The $L_1$ penalty encourages sparsity on the surrogate of direction vector $c$. On the other hand, when solving for $c$, $L_2$ penalty avoids potential singularity in $M$. The $\kappa$ parameter is used to avoid locally optimal solutions and reduce the concavity of the problem. Chun and Keleş [57] note that the problem becomes the PLS original maximum eigenvalue problem when $\kappa = 1$, that of SCOTLASS [60] when original and surrogate direction vectors are equal, i.e., $\alpha = c$, and $M = X^T X$, and that of Sparse Principal Component Analysis (SPCA) [61] when $\kappa = \frac{1}{2}$ and $M = X^T X$. Essentially, the algorithm determines relevant (or active) variables by solving the minimisation problem in Equation (2). Then, using ordinary PLS regression and selected variables, it updates all direction vectors to form a Krylov subsequence—critical for algorithm convergence—on

the subspace of the relevant variables [57]. Previously, this algorithm has been used for crop yield prediction [48], estimation of grass biomass [50], and LAI [49].

The sPLS regression tuning parameters, i.e., $\lambda_1$ (eta) and *K*, were tuned using the 10-fold cross-validation (*cv*) strategy for all the direction vectors. The $\lambda_1$ (or eta) is a sparsity or thresholding parameter and must be between 0 and 1, while *K* is the number of latent (hidden) components and can be between 1 and $min\left\{p, \frac{(v-1)n}{v}\right\}$, where *p* is the number of predictors and *n* is the sample size. The optimal parameters minimised the Mean Squared Error of *CV* (MSE$_{CV}$, Equation (3)). The sPLS analysis was accomplished in R-statistics software [62] with 'spls' package [63]. The MSE$_{CV}$ was converted to Root Mean Squared Error of *CV* (RMSE$_{CV}$) to compare with other MLRAs.

$$\text{MSE}_{CV}\left(\lambda_1, K\right) = \frac{1}{nq} \sum_{s=1}^{10} ||Y_{[s]} - X_{[s]}\hat{\beta}_{\lambda,K}^{(-s)}||_F^2 \tag{3}$$

where $\hat{\beta}_{\lambda,K}^{(-s)}$ is the coefficient estimates with *K* latent components from the *s*th training datasets, i.e., the entire data without *s*th validation dataset, $Y_{[s]}$ and $X_{[s]}$ are the *s*th validation datasets, and $||A||_F^2 = trace\left(A^T A\right)$ is the square of the Frobenius norm.

### 2.4.2. Random Forest

Random Forest (RF) regression [29] is an ensemble tree-based machine learning algorithm and an improvement of Classification and Regression Trees (CART) [64]. In contrast to CART, RF uses bagging (bootstrap aggregating) to iteratively and independently build many decision trees (*ntree*) based on a random subset of training samples created by resampling with replacement from the original sample. Bagging is renowned for its robustness against model overfitting, thus it is critical for obtaining a consistent model [65]. Out of all the training data, about 64% are regarded as in-bag samples, and the remaining 36% are out-of-bag (OOB) samples. These OOB samples are used for evaluating model performance and variable importance [66]. At each node in the tree, a small set of randomly selected features (*mtry*) is used to find the optimal split to grow individual trees. Therefore, the trees grown from different and random subsets ensure increased diversity of decision trees and reduce the bias in the regression model [67]. The final prediction is obtained by aggregating all trees [66]. Variable importance for each explanatory variable, calculated as the Percentage Increase in Mean Squared Error (%IncMSE) when OOB samples are permuted, and while all others are constant, is used to rank the variables according to their predictive power in the model. Essentially, the higher the importance of a specific explanatory variable, the stronger the relationship with the response variable [68]. The optimal *mtry* and *ntree* parameters for estimating LAI, LC$_{ab}$, and CCC with Sentinel-2 MSI were determined by the grid-search strategy [69] using values ranging from 1 to *p* with a single interval, where *p* is the total number of input explanatory variables, and from 100 to 500 with an interval of 10, respectively. Grid-search strategy is a commonly used approach for hyperparameter tuning as it considers, exhaustively, all possible parameter combinations and chooses the pair of parameters that yields minimum OOB error. The RF analysis was performed in R-statistics software [62] using 'randomForest' library [70].

### 2.4.3. Gradient Boosting Machines

Gradient Boosting Machines (GBM), also known as Gradient Boosted Regression Trees (GBRT) [71] uses gradient boosted decision trees and a more regularised formalisation to avoid overfitting, handles missing values (or sparse data) more efficiently, employs parallel and distributed computing for rapid tree construction and building of large models, respectively, and can fit new data added to the trained model. Like RF, it uses CART as a base regressor and the trees are fitted sequentially to minimise the loss function, with new trees added to the model iteratively.

$$\hat{y} = \sum_{k=1}^{K} f_k(x_i) \tag{4}$$

where $\hat{y}$ is the prediction output of sample *i*, $x_i$ is the corresponding input of sample *i*, and $f_k$ is the predictive function of the decision tree *K*. The objective function of the ensemble model is shown in Equation (5).

$$Obj = \sum_{i=1}^{n} l(y_i, \hat{y}_i) + \sum_{k=1}^{K} \Omega(f_k) \tag{5}$$

where $l(y_i, \hat{y}_i) = (y_i - \hat{y}_i)^2$ is the loss function which measures the difference between the predicted $\hat{y}$ and the target $y_i$; $\Omega(f_k) = \gamma T + \frac{1}{2}\lambda \parallel \omega^2 \parallel$ is the regularisation term which represents the complexity of the model; $\gamma$, *T*, $\lambda$, and $\omega$ represent the complexity, the number of leaf nodes in the tree, the fixed coefficient, and the quantised weight vector of leaf nodes, respectively. *K* is the number of trees to be generated and *n* is the number of instances in the training set. A gradient addition strategy is presented by Friedman [71], where new trees are added to the trained model at a time, and the final prediction is achieved through iterative computation.

$$\begin{aligned}
\hat{y}_i^{(0)} &= 0, \\
\hat{y}_i^{(1)} &= f_1(x_i) = \hat{y}_i^{(0)} + f_1(x_i), \\
\hat{y}_i^{(2)} &= f_1(x_i) + f_2(x_i) = \hat{y}_i^{(1)} + f_2(x_i), \\
&\cdots \\
\hat{y}_i^{(t)} &= \sum_{k=1}^{t} f_k(x_i) = \hat{y}_i^{(t-1)} + f_t(x_i)
\end{aligned} \tag{6}$$

where $\hat{y}_i^{(t)}$ and $\hat{y}_i^{(t-1)}$ are the predicted values in the *t*th and $t-1$th rounds, respectively; and $f_t(x_i)$ is the new predictive function of the added tree in the *t*th round. This predictive function and the structure of the new tree is determined through an optimisation algorithm, i.e., greedy algorithm, to minimise the objective function and perform prediction. In this study, GBM was performed using 'gbm' R-package, which required optimisation of four parameters, i.e., the number of trees (n.trees), learning rate (shrinkage), tree depth (interaction.depth), and subsample (bag.fraction) (see Table 2). These were tuned using the grid-search strategy, and the best combination of parameters was selected based on the model fit with the lowest root mean square error of cross-validation ($RMSE_{CV}$). The GBM algorithm also calculates the relative influence of each variable on the model, by averaging the relative influence of variables across all trees.

**Table 2.** Gradient Boosting Machine (GBM) parameters required for optimisation in 'gbm' R-package and their descriptions.

| Parameters | Description |
|---|---|
| Number of trees (*T*) | This is the total number of trees to fit or iterations. |
| Tree depth (*K*) | The depth of a tree determines the number of splits in each tree to control the complexity of the boosted ensemble. |
| Learning rate ($\lambda$) | The learning rate controls the speed of the algorithm down the gradient descent. The smaller values improve the performance and reduce the chance of overfitting. |
| Subsample (*p*) | The subsample ratio of the training instance controls the randomly collected data instance to grow trees. For example, a value of 0.5 causes GBM to randomly collect half of the data instances and prevent overfitting through implementing stochastic gradient descent. The values for this parameter should be between 0 and 1. |

### 2.5. Prediction Accuracy Assessment

The prediction accuracies of each Machine Learning Regression (MLR) model were assessed with the coefficient of determination ($R^2$), root mean squared error (RMSE), and relative root mean squared error (RRMSE) (Equations (7)–(11)). These measures were recommended by Richter, et al. [72] and are frequently used in literature; thus, they facilitate comparison between studies of biophysical parameters. The $R^2$ is a correlation-

based, dimensionless measure that reflects spatial (temporal) patterns, with values $\geq 0.9$ interpreted as Excellent and $0.5 \leq R^2 \leq 0.8$ as Good. In contrast, the RMSE indicates the magnitude of error in the units of the biophysical parameter, i.e., $\text{m}^2 \text{ m}^{-2}$ and $\mu\text{g cm}^{-2}$ for LAI, and $\text{LC}_{ab}$ and CCC, respectively. For LAI, the RMSE values $< 0.5 \text{ m}^2 \text{ m}^{-2}$ can be interpreted as Excellent and $0.5 \text{ m}^2 \text{ m}^{-2} \leq \text{RMSE} < 1.0 \text{ m}^2 \text{ m}^{-2}$ as Good. Lastly, RRMSE is a dimensionless index suitable for comparisons between different variables or ranges, where the values $\leq 10\%$ are regarded as Excellent and $10\% < \text{RRMSE} \leq 20\%$ as Good [72]. Finally, percentage Bias (%Bias) is a measure of the tendency of a model to underestimate or overestimate a biophysical parameter, where the ideal value is 0% and values close to 0% indicate an accurate model [73].

$$R^2 = \frac{\sum (y_i^N - \bar{y}_i)^2}{\sum (y_i - \bar{y}_i)^2} \tag{7}$$

$$\text{RMSE} = \sqrt{\frac{1}{n} \sum_{i=1}^{N} (x_i \times y_i)^2} \tag{8}$$

$$\text{RRMSE} = \frac{\text{RMSE}}{\bar{x}_i} \tag{9}$$

$$\text{MAE} = \frac{1}{n} \sum_{i=1}^{n} |x_i - y_i| \tag{10}$$

$$\%BIAS = \sum_{i=1}^{n} (x_i - y_i) / \sum_{i=1}^{n} (x_i) \tag{11}$$

where $x_i$ is the observed biophysical parameter (e.g., LAI), and $y_i$ is the predicted biophysical parameter (e.g., LAI), $\bar{x}_i$, and $\bar{y}_i$ are the mean of observed and predicted biophysical parameters, respectively; $n$ is the sample size, and $N$ is the number of errors.

All analysis, including training and validation of the models as well as the mapping of biophysical parameters, was performed in R-statistics software [62], while the biophysical parameter maps were generated in ArcMap 10.1 (Environmental Systems Research Institute, Redlands, CA, USA).

## 3. Results

### 3.1. Optimal Tuning Parameters

Table 3 shows the optimal tuning parameters used for the sPLS, RF, and GBM models selected through a grid-search strategy and 10-fold cross-validation. The results are presented for each MLRA and biophysical parameter. sPLS used all variables, $p = 10$, for LAI and CCC, relatively fewer variables, i.e., $p = 7$, are selected for $\text{LC}_{ab}$. The excluded variables for sPLS-$\text{LC}_{ab}$ are B6, B8, and B8A.

**Table 3.** Optimal tuning parameters used to train Leaf Area Index (LAI, $\text{m}^2 \text{ m}^{-2}$), Leaf Chlorophyll Content ($\text{LC}_{ab}$, $\mu\text{g cm}^{-2}$), and Canopy Chlorophyll Content (CCC, $\mu\text{g cm}^{-2}$) models based on grid-search parameterisation strategy and $k$-fold cross-validation.

|  | sPLS | RF | GBM |
|---|---|---|---|
| LAI | eta = 0.7; K = 5; $p = 10$; RMSE$_{CV}$ = 0.8 | m$_{\text{try}}$ = 5; OOB error = 0.34 | n$_{\text{trees}}$ = 146; interaction depth = 5; shrinkage = 0.1; n.minobsinnode = 15; RMSE$_{CV}$ = 0.65 |
| LC$_{ab}$ | eta = 0.9; K = 5; $p = 7$ *; RMSE$_{CV}$ = 7.58 | m$_{\text{try}}$ = 3; OOB error = 7.21 | n$_{\text{trees}}$ = 28; interaction depth = 3; shrinkage = 0.1; n.minobsinnode = 15; RMSE$_{CV}$ = 7.56 |
| CCC | eta = 0.8; K = 5; $p = 10$; RMSE$_{CV}$ = 41.22 | m$_{\text{try}}$ = 2; OOB error = 34.56 | n$_{\text{trees}}$ = 94; interaction depth = 5; shrinkage = 0.1; n.minobsinnode = 15; RMSE$_{CV}$ = 37.21 |

Variables selected by sPLS: * B2, B3, B4, B5, B7, B11, B12.

### 3.2. Model Performance

Sentinel-2 Multi-Spectral Imager (MSI) data were used for estimating LAI, $LC_{ab}$, and CCC using sparse Partial Least Squares (sPLS), Random Forest (RF), and Gradient Boosting Machine (GBM).

The results for LAI (Figure 2a–c) show that RF yields better predictive performance, i.e., RMSE 0.5 $m^2$ $m^{-2}$, explaining 76% of LAI variability, followed by GBM with RMSE of 0.63 $m^2$ $m^{-2}$ and $R^2$ of 0.61, while sPLS is relatively worse, i.e., RMSE: 0.77 $m^2$ $m^{-2}$, and explained only 49% of the variability in LAI. Consistently, sPLS has the greatest %Bias, i.e., 5%, while GBM has the lowest %Bias, i.e., 1%, among the compared MLRAs. Similarly, in predicting $LC_{ab}$ (Figure 2d–f), the results show that the RF algorithm is superior, i.e., RMSE: 7.57 µg $cm^{-2}$, followed by sPLS with RMSE of 7.90 µg $cm^{-2}$, while GBM is relatively worse with 8.25 µg $cm^{-2}$. The variability explained by the sPLS is the lowest (i.e., 81%) when compared to that achieved by RF and GBM models, i.e., 83%, respectively. Moreover, the %Bias is highest in the sPLS model, i.e., 5%, and lowest in the GBM model, i.e., 1%, while RF has a %Bias of 3%. For CCC estimation (Figure 2g–i), RF results show a relatively better predictive performance with RMSE of 39.49 µg $cm^{-2}$, when compared to sPLS and GBM. The GBM model has the next better predictive performance, achieving an RMSE of 44.19 µg $cm^{-2}$, while sPLS is relatively poor with an RMSE of 52.76 µg $cm^{-2}$. Both the GBM-CCC and sPLS-CCC models have markedly high %Bias, i.e., 12% and 13%, as compared to only 2% achieved by the RF-CCC model. Like LAI and $LC_{ab}$ models, the RF-CCC model explains the greatest variability, i.e., 83%, followed by GBM explaining 77% of CCC variability, while sPLS explains the least variability, i.e., 74%, among the compared MLRAs.

### 3.3. Biophysical Parameter Mapping

The three MLRAs, i.e., sPLS, RF, and GBM, were applied to Sentinel-2 MSI image over the study area to characterise the spatial variation of the crop biophysical parameters, i.e., LAI, $LC_{ab}$, and CCC, within and between crop fields. The results are presented in Figure 3. As shown in Figure 3a,d,g, the spatial variation of LAI within the field is discernible with some differences between MLRAs. The sPLS results (Figure 3a) show a wide distribution of low LAI values (i.e., ~2 $m^2$ $m^{-2}$), particularly over rainfed (regular) fields, while the lowest LAI values estimated by RF (Figure 3d) and GBM (Figure 3g) are mostly above 2 $m^2$ $m^{-2}$. However, GBM show relatively lesser values within some fields, indicative of senescing leaves. For $LC_{ab}$ (Figure 3b,e,h), a similar spatial variation can be observed, where sPLS (Figure 3b) results in widely distributed low $LC_{ab}$ values over rainfed fields and maximum values, i.e., ~50 µg $cm^{-2}$, over irrigated fields. In contrast, the $LC_{ab}$ values around 50 µg $cm^{-2}$ are not widely distributed in the $LC_{ab}$ maps estimated from RF (Figure 3e) and GBM (Figure 3h). At the canopy level, the chlorophyll content (i.e., CCC) maps for RF (Figure 3f) and GBM (Figure 3i) are more similar, while sPLS (Figure 3c) contained lower values (i.e., red areas, ~10 µg $cm^{-2}$). The results observed here are consistent with previous studies that found that PLS is well adapted to estimating lower values than other algorithms [74].

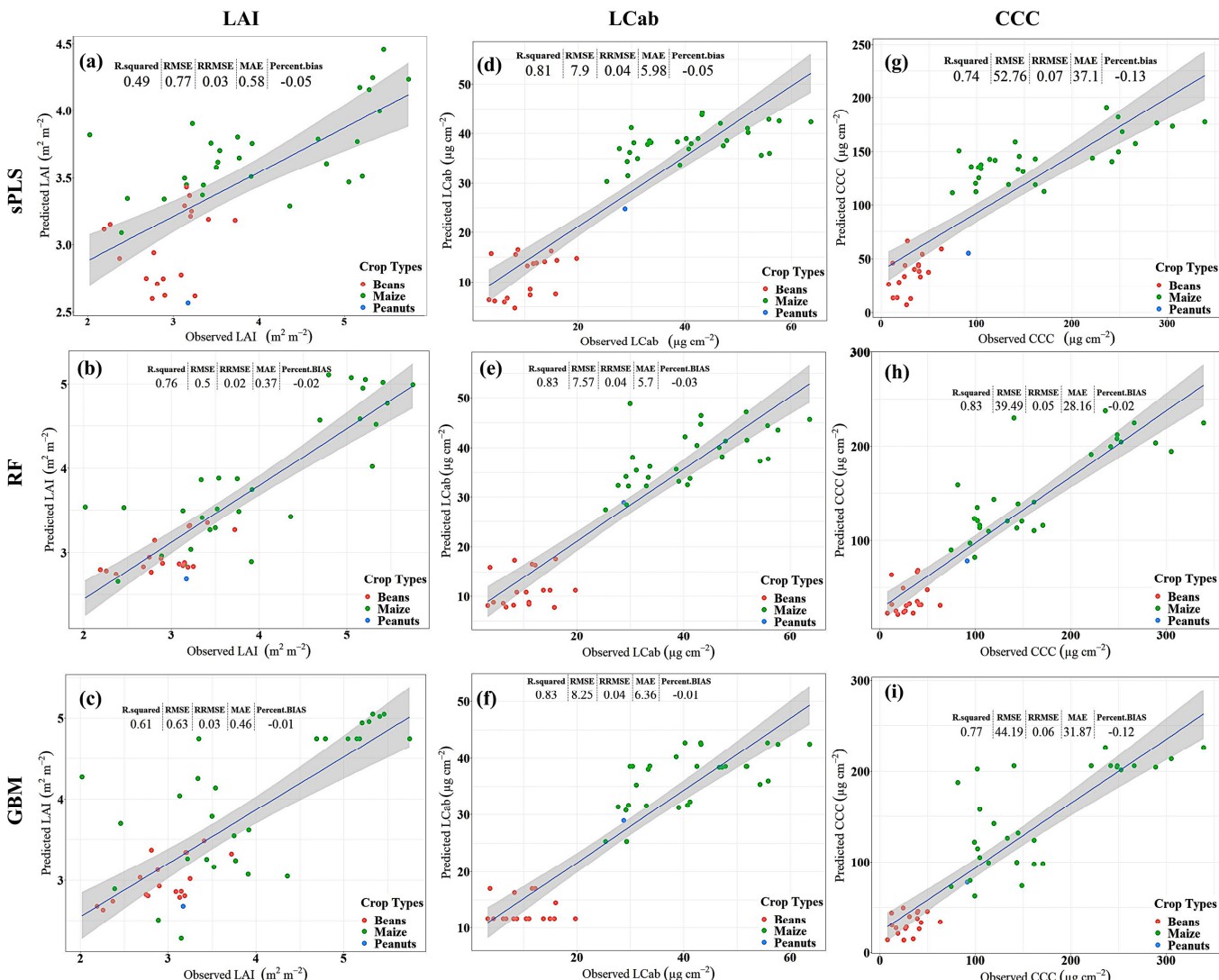

**Figure 2.** Scatterplots for Leaf Area Index (LAI, m$^2$ m$^{-2}$, **a–c**), Leaf Chlorophyll Content (LC$_{ab}$, μg cm$^{-2}$, **d–f**), and Canopy Chlorophyll Content (CCC, μg cm$^{-2}$, **g–i**) showing the performance of sparse Partial Least Squares (sPLS, **a**,**d**,**g**), Random Forest (RF, **b**,**e**,**h**), and Gradient Boosting Machine (GBM, **c**,**f**,**i**) with Sentinel-2 data.

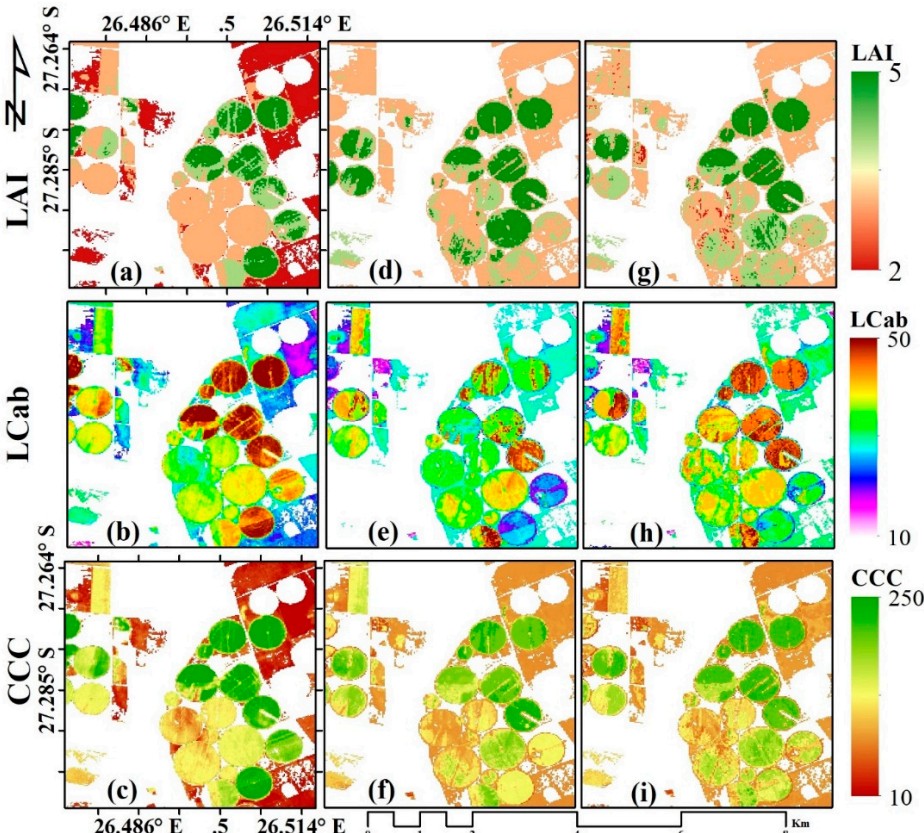

**Figure 3.** Spatial variation of LAI (m$^2$ m$^{-2}$, **a,d,g**), Leaf Chlorophyll Content (LC$_{ab}$, µg cm$^{-2}$, **b,e,h**), and Canopy Chlorophyll Content (CCC, µg cm$^{-2}$, **c,f,i**), estimated with sparse Partial Least Squares (sPLS, **a–c**); Random Forest (RF, **d–f**); Gradient Boosting Machine (GBM, **g–i**) using Sentinel-2 data, respectively.

### 3.4. Variable Importance

Random forest (RF) and Gradient Boosting Machine (GBM) algorithms have built-in variable importance measures, measured by percentage (%) Increase in Mean Square Error (%IncMSE) and Relative influence (%), respectively. For ease of interpreting the results in this study, the spectral bands with importance values, <10%, >10% <20%, and >20%, are regarded as having a low, moderate, and high, predictive power, respectively. Generally, the results (Figure 4) indicate the varying influence of Sentinel-2 spectral bands on the model accuracy for each MLRA and processing level. For the RF-LAI model (See Figure 4a), RE bands, i.e., B5:705 nm, have the highest contribution to the model performance, i.e., >20%, followed by SWIR bands, i.e., B11:1610 nm and B12:2190 nm, and VIS bands, i.e., B3:560 nm, and B4:665 nm, and RE band, B6:740 nm, which have moderate importance, i.e., >10% <20%, while NIR bands, i.e., B8A:865 nm and B8:842 nm, and RE band, i.e., B7:783 nm, have relatively low importance (i.e., <10%). In the GBM model, the VIS band, B4:665 nm, has the highest influence on the model performance, while B11:1610 nm, B5:705 nm, and B3:560 nm, have a moderate influence on the model performance, i.e., >10% <20%. Other bands, i.e., B12:2190 nm, B8A:865 nm, B2:490 nm, B6:740 nm, B7:783 nm, and B8: 842 nm, have the lowest contribution to the model performance, i.e., <10%.

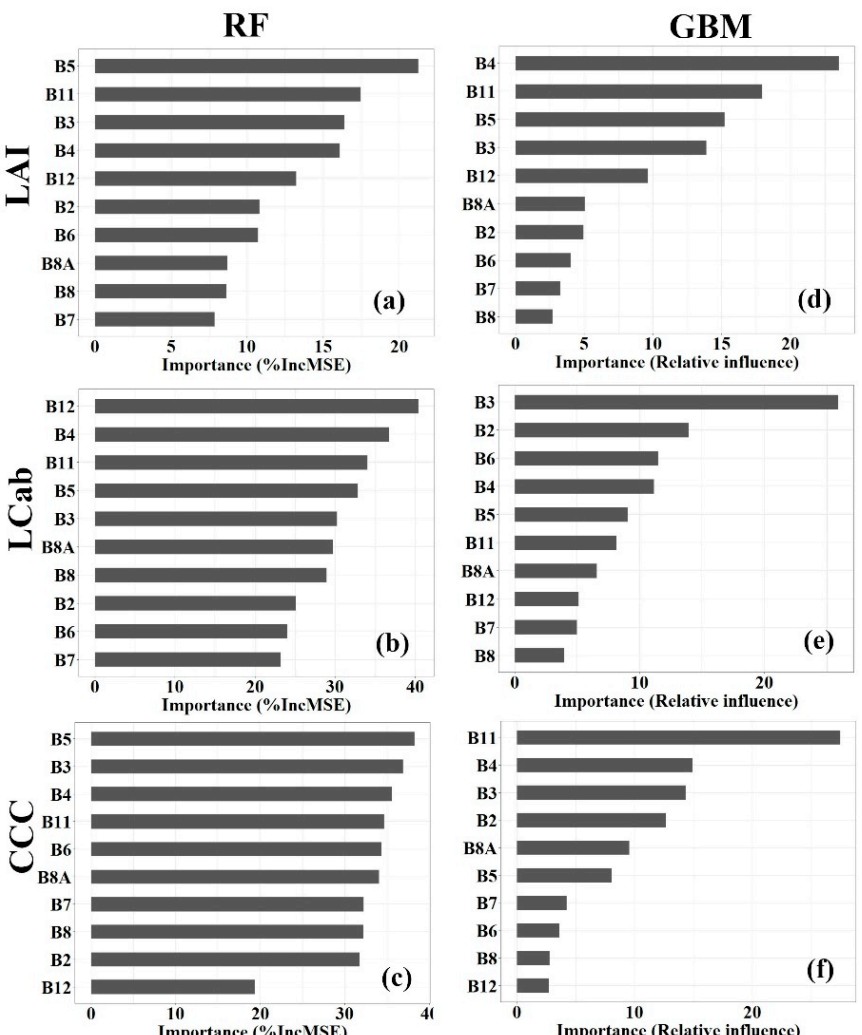

**Figure 4.** Important Sentinel-2 spectral bands (ranked from high to low) for estimating Leaf Area Index (LAI, **a,d**), Leaf Chlorophyll Content (LC$_{ab}$, **b,e**), and Canopy Chlorophyll Content (CCC, **c,f**), from Random Forest (RF, **a–c**) and Gradient Boosting Machine (GBM, **d–f**).

In estimating LC$_{ab}$, the RF importance results (Figure 4b) show that all Sentinel-2 spectral bands have the high contribution the model performance, i.e., >20%, while the GBM model shows that only one variable out of 10 variables (i.e., spectral bands) has a similar high contribution to the model performance, as depicted by Figure 4e. Specifically, the SWIR bands, B11:1610 nm and B12: 2190 nm, RE band, B5:705 nm, and VIS bands, B3:560 nm and B4:665 nm, are the most influential to the RF-LC$_{ab}$ model performance with %IncMSE of >30%, while NIR bands, B8:842 nm, and B8A:865 nm, have relatively low importance alongside VIS band, B2:490 nm, and RE bands, B6:740 nm, and B7:783 nm. In contrast, the GBM results show the greatest influence of the VIS band, B3:560 nm, i.e., >20%, while the VIS bands, B2:490 nm, and B4:665 nm, and RE band, and B6:740 nm, have a moderate influence (i.e., >10% <20%) on the GBM-LC$_{ab}$ model. Like RF-LC$_{ab}$ models, the variable importance for the RF-CCC model (Figure 4c) is also high, i.e., %IncMSE >20%, for all Sentinel-2 spectral bands, except for B12:2190 nm, which has a moderate influence on the model, i.e., >10% <20%. The most important variables, i.e., the VIS bands, B3:560 nm, B2:490 nm, and B4:665 nm, RE bands, B5:705 nm, B6:740 nm, and B7:783 nm, and SWIR band, B11:1610 nm, and NIR band, B8:842 nm, and narrow NIR band, B8A:865, have a %IncMSE of >30%. On the other hand, the GBM-CCC model (Figure 4f) shows that the SWIR band, B11:1610 nm, has the highest influence on the model performance, i.e., >20%, while VIS bands, B4:665 nm, B3:560 nm, and B2:490 nm, have a moderate influence on the

model performance, i.e., >10% <20%. The rest of the other bands, i.e., B8A: 865 nm, B5: 705 nm, B7:783 nm, B6: 740 nm, B8:842 nm, and B12:2190 nm, have the least influence on the model, i.e., <10%.

## 4. Discussion

### 4.1. Predictive Performance of Machine Learning Regression Algorithms

Accurate estimation of crop biophysical parameters from satellite imagery is an ongoing endeavour in various climatic regions across the world due to advances in estimation approaches, satellite data, and changing user needs. In recent times, spatially explicit, site-specific, and accurate biophysical parameters are required at frequent intervals, i.e., 5 days, to optimise farm inputs such as fertilizers, irrigation, and labour and to improve yields and profits. Moreover, international agencies such as United Nations Food and Agriculture Organisation (UN-FAO), regional agencies such as African Union Development Agency (AUDA-NEPAD), and national governments aim to reduce hunger by ensuring adequate and affordable food supply, en-route to achieving the Sustainable Development Goals (UN-SDGs) by 2030. Therefore, ongoing information on the productivity of agricultural systems is highly sought to inform food-related policy- and decision-making. The high-resolution crop biophysical parameters from Sentinel-2 MSI data such as Leaf Area Index (LAI) and chlorophyll content at the leaf (i.e., $LC_{ab}$) and canopy levels (i.e., CCC) are essential for both precision agriculture and crop monitoring needs. This paper evaluated the performance of three machine learning regression algorithms (MLRAs) in estimating these foliar biophysical parameters in an African semi-arid agricultural area in Bothaville, South Africa.

Machine Learning Regression Algorithms (MLRAs) are among the most promising biophysical retrieval approaches and have the greatest potential for improving the accuracies and reliability of biophysical parameters from satellite data. One of the attractive features of MLRAs is their capability to determine non-linear relationships between biophysical parameters of interest and satellite-based co-variates such as spectral reflectance. The results in this study (Figure 2) demonstrated that the RF regression algorithm yielded superior predictive performance in estimating LAI, $LC_{ab}$, and CCC across compared MLRAs. These results present the RF as a good contender for the operationalization of biophysical parameters to support precision agriculture and crop monitoring since all relevant parameters for detecting crop health could be achieved in a single algorithm. Previous MLRA comparison studies found that different algorithms were optimal for characterizing different and individual parameters [17,75]. Therefore, the results here are significant for product developers aiming to reduce reliance on multiple algorithms for multiple parameter estimation. The algorithm's inner workings are one of its greatest strengths as they are relatively transparent and require a few and intuitive parameters, i.e., $m_{try}$ and $n_{tree}$, when compared to complex algorithms such as NN. For example, since it is based on the traditional CART algorithm, one can interrogate the variables used to split the nodes in the individual trees and thus explain the predictions. The strength of ensemble learning methods is also shown by the better predictive performance of GBM in estimating LAI and CCC. Although RF and GBM use CART as a base regressor, GBM makes predictions by sequentially and iteratively combining weak regression trees to improve the predictive performance rather than independently constructing several decision trees [76]. However, as shown by the results in this study, the RF formulation is more superior in terms of predictive performance.

The weakness of the GBM formulation is also shown by its relatively worse performance when estimating $LC_{ab}$, compared to sPLS which showed relatively better prediction accuracy. The relatively poor performance of sPLS in estimating other biophysical parameters, i.e., LAI and CCC, can be attributed to its inability to deal with non-linear relationships between explanatory variables, i.e., spectral bands, or between these variables and the biophysical parameters [74]. Since Sentinel-2 contains optimized and few spectral bands for land monitoring, the variable selection capability of sPLS was rather not fully exploited.

Thus, the observed results can be attributed to ordinary PLS used in the algorithm for projecting the variables into orthogonal space and estimation of biophysical parameters, with the exception of the sPLS-LC$_{ab}$ model. Concerning the sPLS-LC$_{ab}$ model, the capability of reducing dimensionality is observed, which assisted its better prediction accuracy against GBM. Despite its better performance in terms of RMSE and $R^2$, it showed greater underestimations, i.e., %Bias ~5%, when compared to GBM. Delloye, et al. [77] found that extreme LC$_{ab}$ values, i.e., <30 and >70 μg cm$^{-2}$, tend to be under-estimated. Our results showed this underestimation at >50 μg cm$^{-2}$, possibly due to the difference in the range of training and validation data and the limitations of MLRAs in estimating beyond the range of training data. In fact, GBM resulted in the lowest %Bias for LAI and LC$_{ab}$ compared to both RF and sPLS. Therefore, its potential should be explored further in future studies, particularly its improved version, i.e., eXtreme Gradient Boosting (XGB), which has been reported highly efficient and more accurate.

The predictive performance of RF in estimating LAI found in this study (RMSE: 0.5 m$^2$ m$^{-2}$, $R^2$: 0.76) is slightly low compared to that found by Verrelst, et al. [17], i.e., RMSE$_{CV}$:0.44 m$^2$ m$^{-2}$ ($R^2$: 0.90), using variational heteroscedastic Gaussian Process regression (VHGPR) and simulated Sentinel-2 bands in Barrax, Spain, though their data had a higher spatial resolution, i.e., 5 m. Over the same site, Verreslt, et al. [20] found an RMSE of 0.49 m$^2$ m$^{-2}$ ($R^2$: 0.92) using Gaussian Process regression (GPR) and simulated Sentinel-2 data at 20 m resolution. This finding is comparable to the RF-LAI results in this study; however, their RRMSE is considerably worse, i.e., 23%, thus failing to reach the prescribed minimum accuracy of 10% for LAI product [78]. Our LC$_{ab}$ prediction performance was slightly worse than that reported by Delloye, Weiss and Defourny [77] using NN and Sentinel-2 data, i.e., RMSE: 7.26 μg m$^{-2}$ and better than that reported by Upreti, et al. [75], i.e., RMSE: 8.88 μg cm$^{-2}$ using Sentinel-2 data and RF. Considering that all the MLRAs evaluated in this study resulted in considerably better RRMSE, i.e., 2–3%, the study supports the recommendation by [17] that slower and non-optimal MLRAs such as NN should be replaced by simpler but more powerful ones.

*4.2. Influential Variables for Biophysical Parameter Estimation*

The measures of variable importance have been used extensively in the literature to explain the performance of the modelled relationships with biophysical parameters, as well as variable selection [51,68,79]. Moreover, they probe the structure and parameters learned by a trained MLR model to understand important variables or combinations of important variables for deriving predictions. Fortunately, the MLRAs used here, i.e., RF and GBM, had built-in variable importance measures, while the selected variables by sPLS can be considered to contain high information content. The variable importance results (Figure 4) showed varying importance for each biophysical parameter and MLRA. This is not surprising since the different biophysical parameters affect various regions of the electromagnetic spectrum differently. However, their influence is not exclusive to particular regions. For example, chlorophyll is known to have strong absorption in the 400–500 nm region (blue) and 650–700 nm region (red-edge), while carotenoids also absorb the blue radiation. Moreover, the water content in leaves causes strong absorption features in the SWIR region (1450 nm and 1950 nm) and relatively weak absorption in the NIR region (i.e., 980 nm and 1150 nm) [14]. Both regions have been associated with an accurate estimation of biophysical parameters such as LAI since most plant functions depend on water [17]. In agreement, our results showed that the high influence of SWIR, VIS, and RE bands was evident over NIR bands, which were less important for LAI using RF and all biophysical parameters using GBM.

Among the three Sentinel-2 MSI RE bands, B5 centred at 705 nm was more important in estimating all the biophysical parameters considered here, irrespective of the algorithm used. Wu, et al., [80] showed that spectral vegetation indices using this band (i.e., 705 nm) are better estimators of chlorophyll content than other RE bands. This finding is consistent with Verrelst, et al. [81] who found that wavelengths above 730 nm resulted in poor LC$_{ab}$

results due to low absorption of solar radiation by chlorophyll pigments. As a result, $LC_{ab}$ induces the largest variation in reflectance in the RE region below 730 nm [77]. The other Sentinel-2 RE bands are centred above 730 nm, i.e., B6:740 nm and B7:783 nm. Interestingly, both LAI and CCC models were mostly influenced by somewhat similar spectral bands. For example, in addition to B5, B11:1610 nm, B3:560 nm, and B4:665 nm, showed the greatest influence on the MLR model performance. It is, therefore, clear that these bands have high information content, explaining the greatest variability of LAI, $LC_{ab}$, and CCC in the study area. The similarity in variable importance between LAI and CCC reflects the co-variation of LAI and $LC_{ab}$ at the canopy level. The VIS bands, B3:560 nm, and B4:665 nm, and SWIR band, B11:1610 nm, featured prominently in $LC_{ab}$ models, while the contribution of NIR was relatively low. Consistently, the sPLS-$LC_{ab}$ model excluded NIR bands, B8, and B8A. In rare instances, i.e., $LC_{ab}$ and CCC with RF, NIR bands, B8:842 nm and B8A:865 nm, were among the highly influential variables (i.e., %IncMSE >20%) to the model performance. Nonetheless, it should be noted that all the Sentinel-2 MSI co-variates contributed towards the MLRA performance, hence they should not be discarded. This is especially true for LAI which has been found to drive spectral variation in all bands and due to co-variation of biophysical parameters in various spectral regions [13,14]. However, it may be interesting to test the different subsets consisting of four to 10 bands based on these variable importance measures. This suggestion is informed by the sPLS-$LC_{ab}$ model which used fewer, i.e., 7 variables, resulting in better performance over GBM. Consistently, Verrelst, et al., [13] found that fewer HyMap bands, i.e., 9 and 7, were optimal ($NRMSE_{CV}$: <10%) for estimating $LC_{ab}$ and LAI, respectively. In another study, Verrelst, et al., [81] found that only four Compact High-Resolution Imaging Spectrometer (CHRIS) bands, i.e., 674 nm, 605 nm, 942 nm, and 978 nm and 725 nm, 471 nm, 997 nm, and 511 nm, are sufficient to obtain accurate LAI and $LC_{ab}$ estimations, respectively. Therefore, the RF and GBM importance measures can be explored in future studies to optimise the number of spectral bands for estimating biophysical parameters with MLRAs based on some threshold.

As shown by the results, RF seems to utilize the information content of all Sentinel-2 bands more efficiently than the GBM, where all variables had relatively higher or close importance values, especially for $LC_{ab}$ and CCC (Figure 4b,c). In contrast, GBM's most important variable showed a markedly high relative influence compared to the next or other important variables. This observation can be attributed to the differences in the architecture of these MLRAs (i.e., RF and GBM). RF selects a subset of random variables to split the nodes of each tree; thus, all Sentinel-2 bands have an equal chance of contributing to the model. This may work against or in favour of prediction performance. For example, in datasets containing highly correlated variables, such as hyperspectral data, collinear variables may be selected for all or most of the trees in the forest, resulting in overfitting [82]. In such a case, feature selection and dimensionality reduction become essential [13,74]. Fortunately, in our case, the spectral bands were discrete and any redundant bands, such as B12:2190 nm, B6:740 nm, B7:783 nm, B8:842 nm, and B8A:865 nm, ranked relatively low in variable importance. The influential variables found here are consistent with the known absorption features and relationships with plant biophysical parameters [11,12,14].

### 4.3. Limitations of the Study

Despite the better performance of RF in this study, the model was not tested in a different study area to confirm its prediction consistency and transferability. In fact, transferability is one of the key issues in biophysical parameter estimation using MLRAs due to the high cost of collecting calibration data. The coupling of RTM-MLRA seems to be a promising solution for operationalisation, despite the ill-posed nature of physically-based approaches. Unfortunately, such efforts have focused mainly on NN [77,83], which has been proven inconsistent in semi-arid African environments [37]. Moreover, variable importance used to explain the model performance here is inadequate as it provides global interpretations (i.e., based on the entire model architecture and dataset) and explains overall relationships between the explanatory and response variables. However, it is desirable to

understand the local explanations of the model, i.e., why a specific prediction was made for a particular observation. In the future, we will explore the transferability and local explanations of MLRAs. Finally, the MLRAs predictions were limited to the dynamic range of input field data. Nonetheless, the data represented the crop conditions at a specific time. Moreover, the possible errors in the field data measured using field instruments were propagated to the MLRAs [7]. Other hyperparameter tuning strategies should be explored in the future in addition to grid search strategy to evaluate their consistency and effect on prediction accuracy. It is further recommended to validate (with field reflectance data) the surface reflectance data derived from various atmospheric correction approaches (including Sen2Cor used here) as well as TOA reflectance data to not only inform the selection of the most accurate input bands for estimating crop biophysical parameters but also further our understanding of the effect of various AC approaches on such parameters as recommended by Djamai and Fernandes [84]. It is anticipated that improved atmospheric correction algorithms may yield better results and future studies should explore this.

## 5. Conclusions

This paper evaluated the performance of sparse Partial Least Squares (sPLS), Random Forest (RF), and Gradient Boosting Machine (GBM) in estimating LAI, $LC_{ab}$, and CCC using Sentinel-2 data. Moreover, the spectral bands that had greatest influence on the model accuracy were identified using RF and GBM variable importance measures. The results showed that RF was superior in estimating all three biophysical parameters, followed by GBM which was better than sPLS in estimating LAI and CCC but not $LC_{ab}$, where sPLS showed relatively better prediction accuracy. Nevertheless, all MLRAs resulted in acceptable accuracy by GCOS/GMES, i.e., RRMSE of $\leq$10%, for all the biophysical parameters. This result is comparable (in other cases better when compared) to studies using simulated and hyperspectral data, RTMs, and advanced MLRAs such as NN and GPR [17,20,77,83]. Based on sPLS' better predictive performance using only seven variables over GBM in estimating $LC_{ab}$, it is recommended that fewer, i.e., 4–8, Sentinel-2 spectral subsets should be evaluated in the future. This recommendation is consistent with previous studies [13,81] that found that four to nine bands were sufficient to achieve robust estimates of biophysical parameters. Among all Sentinel-2 MSI bands, B5 (705 nm) was consistently more important in estimating all the biophysical parameters considered here. This band was previously shown to be a better estimator of chlorophyll content [80,81] than other RE regions. Interestingly, the results also showed some level of dependency between canopy biophysical parameters, i.e., LAI and CCC, as the variables with the greatest influence on the model performance were somewhat similar, i.e., B5, B11, B3, and B4. This finding shows that these bands have high information content, hence high predictive power, and are highly related to both LAI and CCC; this is consistent with previous studies that found that structural and leaf parameters have a co-dependent effect on canopy spectral variations which is difficult to decouple [14,77]. Overall, our results demonstrated that the RF regression algorithm provides the most robust predictive performance for all the crop biophysical parameters considered here by efficiently utilising all Sentinel-2 MSI bands, thus it is a good contender for operationalisation. However, this assertion should be tested further in different growth stages, crops, and climatic environments, in future studies. This is essential for future satellite-based biophysical product development using a single MLRA, thus improving the prospects and effectiveness of developing accurate prescription maps to support VRA precision agriculture techniques and regional crop monitoring activities.

**Author Contributions:** Conceptualization, M.K.; methodology, M.K.; formal analysis, M.K.; writing—original draft preparation, M.K.; writing—review and editing, M.K., C.A. and P.M.; visualization, M.K.; supervision, C.A. and P.M.; All authors have read and agreed to the published version of the manuscript.

**Funding:** This research was supported by the AfriCultuReS project, which received funding from the European Union's Horizon 2020 Research and Innovation Framework Programme under grant agreement No. 774652. Mahlatse Kganyago received European Space Agency (ESA) Network of Resources (NoR) sponsorship for Sentinel Hub (by Synergise) subscription. Mahlatse Kganyago received Postgraduate Merit Award (PMA) from the University of the Witwatersrand. The APC was sponsored by Prof. Paidamwoyo Mhangara (University of the Witwatersrand).

**Acknowledgments:** The authors appreciate the field data provided by the EU-H2020 AfriCultuReS project (GA: 774652), the ESA Network of Resources (NoR) sponsorship for providing access to Sentinel Hub Cloud API for Satellite Imagery used in this study, and the University of the Witwatersrand for the Postgraduate Merit Award (PMA). We appreciate the support provided by the South African National Space Agency (SANSA). The following SANSA individuals are highly appreciated for taking part in the 2021 fieldwork in Bothaville alongside the main author: Nosiseko Mashiyi Thomas Tsoeleng, and Morwapula Mashalane. Finally, we thank anonymous reviewers and editor(s) for taking the time to provide constructive feedback on this manuscript.

**Conflicts of Interest:** The authors declare no conflict of interest.

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
