# Peer review of "Estimating Crop Biophysical Parameters Using Machine Learning Algorithms and Sentinel-2 Imagery"

_remotesensing, doi:10.3390/rs13214314_

Round 1

Reviewer 1 Report

The paper titled "Estimating Crop Biophysical Parameters using Machine Learning Algorithms and Sentinel-2 Imagery" was well written with main research of estimating biophysical parameters of crop using machine learning approaches. 

-The introduction section was well written by introducing the importance of precision agriculture, and significance of remote sensing satellite images in estimating biophysical parameters of crops.

- The study area was clearly described with geographical location map, climatic conditions and crop seasons.

- Remote sensing Sentinel-2 MSI spectral characteristics and Calibration & validation for LAI, LCab and CCC were discussed very clearly.

- the results and discussions were satisfied the objectives of the paper and authors were presented the results.

- The concluding remarks were acceptable.

-Over all the paper has unique research and, the authors were well organized and presented the article.

Author Response

We thank the reviewer for taking their time to review the manuscript and providing comments.

Reviewer 2 Report

Vegetation biophysical and chemical products are of great significance to agriculture and forestry. Therefore, it is still meaningful to carry out vegetation parameter estimation, especially by using satellite data. This paper attempts to compare the effectiveness of three common machine learning methods (not simple regression), and tries to test with TOA data, trying to prove its effectiveness in vegetation parameter estimation. The language of the paper is good. However, the experimental methods in this article have the lack of innovation, this leads to a lack of sufficient and more scientific argumentation for some conclusions. It is recommended to reorganize the paper.

There are two big issues to consider in order to make the new paper better.
(1) This paper has got some credible conclusions of the comparison of three common machine learning methods in estimating LAI, LCab, and CCC. The three machine learning regression methods tested in the paper are relatively normal and have been tested in many studies for vegeation parameter estimation. Although the application of several common used MLRA methods in this paper is a little valuable, it does not constitute the innovation of this paper.
(2) Atmospheric correction has been considerd as an important work for vegetation parameter estimation, it is necessary to explore the vegetation parameter estimation without atmospheric correction. However, the experimental methods and data in this paper are difficult to prove that the established model has application potential. The single-scene sentinel-2 satellite imagery under clear sky conditions cannot reflect the influence of the complex atmospheric condtions on the vegeation parameter estimation. The deviation of remote sensing signal caused by the atmosphere in the single-scene imagery on a sunny day is relatively consistent. Using machine learning methods or even simple regression, TOA and TOC are likely to have similar performance of vegeation parameter estimation. In addition, the machine learning model based on TOA data on one day cannot be extended to another day because the atmospheric conditions are very likely to be different. But the TOC-based model has advantages. This article needs to get new innovation points.

Author Response

We appreciate the reviewer's comments and their time in reviewing the manuscript.

Reviewer’s comments:

There are two big issues to consider in order to make the new paper better.
(1) This paper has got some credible conclusions of the comparison of three common machine learning methods in estimating LAI, LCab, and CCC. The three machine learning regression methods tested in the paper are relatively normal and have been tested in many studies for vegeation parameter estimation. Although the application of several common used MLRA methods in this paper is a little valuable, it does not constitute the innovation of this paper.

  • We take note of the reviewer’s concern, but we disagree that the paper makes no new contributions. First, we did not find any study, published in the past decade that systematically compared the performance of these particular algorithms, especially in the context of estimating all the crop biophysical parameters considered in the current study. Therefore, the unique contribution here is that the study shows how these algorithms differ in their performance in a semi-arid agricultural area and under the same prevailing conditions. Second, the criteria we used for the choice of these algorithms is their simplicity, robustness and popularity in several remote sensing applications domains. Moreover, they require lesser and intuitive tuning parameters and are explainable as compared to kernel-based and deep learning algorithms such as Kernel Ridge Regression (KRR), Support Vector Machines (SVM) and Neural Networks (NN), which are “black-boxes”, complex and computationally expensive. The motivation for using these algorithms has been added to the manuscript.

(2) Atmospheric correction has been considerd as an important work for vegetation parameter estimation, it is necessary to explore the vegetation parameter estimation without atmospheric correction. However, the experimental methods and data in this paper are difficult to prove that the established model has application potential. The single-scene sentinel-2 satellite imagery under clear sky conditions cannot reflect the influence of the complex atmospheric condtions on the vegeation parameter estimation. The deviation of remote sensing signal caused by the atmosphere in the single-scene imagery on a sunny day is relatively consistent. Using machine learning methods or even simple regression, TOA and TOC are likely to have similar performance of vegeation parameter estimation. In addition, the machine learning model based on TOA data on one day cannot be extended to another day because the atmospheric conditions are very likely to be different. But the TOC-based model has advantages.

  • We have now removed all the TOA analyses.

Reviewer 3 Report

Dear authors,

It is very good article and thanks for you work and contributions.

  1. Why the validation data is only 30%?
  2. Please can you add some explanations about the software used for this research. Did you apply R software (line #342 for all the methods?
  3. Please can you add some explanations why the methods are selected for this study (RF, PLS, GBM).

Author Response

We appreciate the reviewer for taking the time to provide this valuable feedback. Below, we respond to the reviewer’s comments. We hope these will be satisfactory to warrant acceptance for publication. 

1. Why the validation data is only 30%?

  • The split of data into 70% training vs. 30% validation is common in machine learning regression and classification, to ensure robust and unbiased predictions and a sufficient amount, commonly 30%, of these samples is kept aside to perform independent validation (https://doi.org/10.1080/10106049.2014.997303; https://doi.org/10.1371/journal.pone.0224365; https://doi.org/10.1371/journal.pone.0249136).

Studies such as http://dx.doi.org/10.1016/j.isprsjprs.2013.09.012  and https://doi.org/10.1109/TGRS.2011.2168962  regard 20% of the samples as adequate to “…employ a solid validation”. 

We have added a reference to the sentence in the revised manuscript, to support our choice of 70% training vs. 30% validation.

2. Please can you add some explanations about the software used for this research. Did you apply R software (line #342 for all the methods?

  • Yes, all the analysis was performed in R Statistics, while the biophysical parameter maps were generated in ArcMap 10.1 (ESRI, Redlands, CA). We have now clarified this in the revised manuscript.

3. Please can you add some explanations why the methods are selected for this study (RF, PLS, GBM).

Explanations motivating our choice of the three algorithms were added in subsection 2.4. of the revised manuscript as per the reviewer’s suggestion. 

Reviewer 4 Report

Comments for remotesensing-1388402

General comments:

The authors submitted the manuscript, which presents the results of their work in a proper way, according to the high standards applied for scientific publications. The authors firstly explored the biophysical parameters simulations using three machine learning algorithms based on TOA and BOA data from Sentinel-2 imagery. The models were calibrated and evaluated well. And the performance of the three models was compared to previous studies.  Conclusions were made according to findings.

I think this manuscript has an original design and enough experiments to support the conclusions. However, I still have some concerns about the manuscript’s quality. Firstly, please unify the tense in the introduction section. Some of them used past tense, while some used present tense when the authors described previous studies. Secondly, please unify the capitalized terminologies, such as Random Forest, Top-of-Atmosphere. In addition, there are many small mistakes or typos in the manuscript. Hope the authors can carefully correct them.

As regards some specific suggestions, see the followed specific comments.

I recommend publication after careful revisions.

Specific comments:

  1. Line 19: add “,” after “precision irrigation”.
  2. Line 58: revise “show” to “shows”.
  3. Line 60: add “and” before “residual nitrates”.
  4. Line 72: add “,” after “health”.
  5. Line 122: add “,” after “climate zones”.
  6. Line 130: add “,” after “complex”.
  7. Line 134: add “,” after “support vector machines (SVM)”.
  8. Lines 143 and 145: revise “show” to “showed”.
  9. Lines 160 and 163: add “,” after “LCab”.
  10. Line 168: the authors mention two experimental sites, but only give one. Why did not mention the other one?
  11. In figure 1, are these two images acquired on the same date? In addition, please clarify the scale belonging to which panel,  adds the direction arrow in the figure.
  12. Line 130: add “,” after “snow”.
  13. Lines 220-222: field collection used 40m*40m plots, while the images have 20 m spatial resolution. How does the author match the field observations and image pixels?
  14. Lines 230-231: for the calculation of CCC, if the authors can provide some references?
  15. Lines 238-239: please revise the title of Table 1. Revise “dataset” to “datasets”
  16. Line 253: please improve equation 1, delete the “=” marks.
  17. Lines 366-367: in equations 7 and 8, there is no explanation about the N value. Is it the same with n?
  18. Line 377: Table 3 “show” change to “shows”
  19. Line 390: add “,” before “and”
  20. Line 396: add “,” before “and”
  21. Lines 416, 431, and 447: add “,” before “and”
  22. Line 414: Figure 2: please the fond of words in each graph. And align the two lines in (c) and (f).

In addition, the statistics in each growth were calculated from three crops or a single one?

  1. Line 417: “show” change to “shows”
  2. Lines 435-436: revise this sentence.
  3. In Figures 3 and 4: why the R2 can explain the variability? Does it represent the feature importance generated from three models?
  4. Line 462: add respectively at the end of the sentence
  5. In Figures 5 - 7, add the legend title and unit.
  6. In Figures 8 -10, they make readers confused. please clarify which graph is for TOA and which one is for BOA.
  7. Line 577: Figures 2-4
  8. Lines 596-600: Here, the authors present the worse performance of GBM in estimating Lcab. However, they did not explain the reason. Instead, the reason for the poor performance of sPLS was provided. It was confusing and contradictory.
  9. Line 577: Figures 8-10

Author Response

We highly appreciate the reviewer for taking the time to review the paper and provide constructive comments.

General comments:

The authors submitted the manuscript, which presents the results of their work in a proper way, according to the high standards applied for scientific publications. The authors firstly explored the biophysical parameters simulations using three machine learning algorithms based on TOA and BOA data from Sentinel-2 imagery. The models were calibrated and evaluated well. And the performance of the three models was compared to previous studies.  Conclusions were made according to findings.

I think this manuscript has an original design and enough experiments to support the conclusions.

 However, I still have some concerns about the manuscript’s quality. Firstly, please unify the tense in the introduction section. Some of them used past tense, while some used present tense when the authors described previous studies.

  • We have now ensured correct and consistent tense usage throughout the manuscript.

Secondly, please unify the capitalized terminologies, such as Random Forest, Top-of-Atmosphere. In addition, there are many small mistakes or typos in the manuscript. Hope the authors can carefully correct them.

  • The capitalization has now been unified as per the reviewer’s suggestion.

As regards some specific suggestions, see the followed specific comments.

I recommend publication after careful revisions.

Specific comments:

1. Line 19: add “,” after “precision irrigation”.

  • Comma added.

2. Line 58: revise “show” to “shows”.

  • We have revised the way the reference is written. It now reads, Stamatiadis, et al. [5] show...
  • 3. Line 60: add “and” before “residual nitrates”.
  • “and” was added.

4. Line 72: add “,” after “health”.

  • “and” was added.

5. Line 122: add “,” after “climate zones”.

  • Comma added.

6. Line 130: add “,” after “complex”.

  • Comma added.

7. Line 134: add “,” after “support vector machines (SVM)”.

  • Comma added.

8. Lines 143 and 145: revise “show” to “showed”.

  • We have now ensured correct and consistent tense usage throughout the manuscript. Where a previous study is referred, we used present tense as it regarded existing knowledge, while our discussion is written in the past tense.

9. Lines 160 and 163: add “,” after “LCab”.

  • Comma added.

10. Line 168: the authors mention two experimental sites, but only give one. Why did not mention the other one?

  • This was a mistake, we have now corrected this.

11. In figure 1, are these two images acquired on the same date? In addition, please clarify the scale belonging to which panel,  adds the direction arrow in the figure.

  • The scale is for both images. We have now included the dates in the caption.

13. Line 130: add “,” after “snow”.

  • Comma was added.

14. Lines 220-222: field collection used 40m*40m plots, while the images have 20 m spatial resolution. How does the author match the field observations and image pixels?

  • We have extracted the pixel-values that intersected with the 40m by 40m blocks from the Sentinel-2 image. We have now clarified this in the manuscript.

15. Lines 230-231: for the calculation of CCC, if the authors can provide some references?

  • We have now provided a reference as recommended.

15. Lines 238-239: please revise the title of Table 1. Revise “dataset” to “datasets”

  • Table heading is now revised as recommended.

16. Line 253: please improve equation 1, delete the “=” marks.

  • The “=” marks indicate that the respective thresholds were assigned class “0” and “1”, respectively. We do not understand how their removal would improve the equation.

17. Lines 366-367: in equations 7 and 8, there is no explanation about the N value. Is it the same with n?

  • We have now described all the equation terms.

18. Line 377: Table 3 “show” change to “shows”

  • Change has been effected.

19. Line 390: add “,” before “and”

  • Comma added.

20. Line 396: add “,” before “and”

  • Comma added.

21. Lines 416, 431, and 447: add “,” before “and”

  • Comma added on the respective lines.

22. Line 414: Figure 2: please the fond of words in each graph. And align the two lines in (c) and (f).

  • The two lines are now aligned.

In addition, the statistics in each growth were calculated from three crops or a single one?

  • The statistics are for the combined crops, we have clarified this in the revised manuscript.

23. Line 417: “show” change to “shows”

  • Change has been effected,

24. Lines 435-436: revise this sentence.

  •  

25. In Figures 3 and 4: why the R2 can explain the variability? Does it represent the feature importance generated from three models?

  • R-squared (R2) is a statistical measure that represents the proportion of the variance for a dependent variable that's explained by an independent variable or variables in a regression model. R-squared explains to what extent the variance of one or more variables explains the variance of the second variable. So, if the R2 of a model is 0.50, then approximately half of the observed variation can be explained by the model's inputs.
  1. Line 462: add respectively at the end of the sentence
  • The suggested change has been effected.

27. In Figures 5 - 7, add the legend title and unit.

  • The units for each biophysical parameter were added caption of the figures and legend titles were also added.

28. In Figures 8 -10, they make readers confused. please clarify which graph is for TOA and which one is for BOA.

  • We appreciate the reviewer’s suggestion. We have clarified this in the captions of the mentioned figures.

29. Line 577: Figures 2-4

  • The suggested change has been effected.

30. Lines 596-600: Here, the authors present the worse performance of GBM in estimating Lcab. However, they did not explain the reason. Instead, the reason for the poor performance of sPLS was provided. It was confusing and contradictory.

  • We have now revised the sentence.

31. Line 577: Figures 8-10

  • The suggested change has been effected.

Round 2

Reviewer 2 Report

The author has made major revisions to the comments of the previous version, especially the part of TOA has been deleted  in this paper, and the TOC part has been retained. I think the TOC part of the research can become a complete paper. According to the current revision quality of the manuscript, this paper can be published.